# The Phosphorylated Form of the Histone H2AX (γH2AX) in the Brain from Embryonic Life to Old Age

**DOI:** 10.3390/molecules26237198

**Published:** 2021-11-27

**Authors:** Adalberto Merighi, Nadia Gionchiglia, Alberto Granato, Laura Lossi

**Affiliations:** Department of Veterinary Sciences, University of Turin, I-10095 Grugliasco, Italy; adalberto.merighi@unito.it (A.M.); nadia.gionchiglia@unito.it (N.G.); alberto.granato@unito.it (A.G.)

**Keywords:** H2AX, DNA damage, neurons, neurogenesis, apoptosis, aging, mitosis, cerebral cortex, subventricular zone

## Abstract

The γ phosphorylated form of the histone H2AX (γH2AX) was described more than 40 years ago and it was demonstrated that phosphorylation of H2AX was one of the first cellular responses to DNA damage. Since then, γH2AX has been implicated in diverse cellular functions in normal and pathological cells. In the first part of this review, we will briefly describe the intervention of H2AX in the DNA damage response (DDR) and its role in some pivotal cellular events, such as regulation of cell cycle checkpoints, genomic instability, cell growth, mitosis, embryogenesis, and apoptosis. Then, in the main part of this contribution, we will discuss the involvement of γH2AX in the normal and pathological central nervous system, with particular attention to the differences in the DDR between immature and mature neurons, and to the significance of H2AX phosphorylation in neurogenesis and neuronal cell death. The emerging picture is that H2AX is a pleiotropic molecule with an array of yet not fully understood functions in the brain, from embryonic life to old age.

## 1. Introduction

The variant H2AX of the histone H2A was originally described by West and Bonner in 1980 [1]. Thereafter, it was demonstrated to represent 2.5–25% of H2A in the total mammalian genome and to be specifically phosphorylated in response to DNA damage [2]. Initially, phosphorylation of H2AX (referred to as γH2AX form) because of the formation of DNA double-strand breaks (DSBs) was considered the archetypal role of H2AX. However, more recently, it became clear that H2AX has several other “non-canonical” functions during the normal life of the cell and in certain pathological conditions [3,4]. Yet, the involvement of γH2AX in the normal and pathological brain has received only a little consideration. In this review, we will discuss the literature on γH2AX distribution and function in the brain. We will pay particular attention to its relationship with neuronal cell death and neurodegeneration.

### 1.1. The Histone Families, H2A, and the DNA Damage Response

Five families of histones—H1, H2A, H2B, H3, and H4—organize and package eukaryotic DNA to form a nucleic acid-protein complex called chromatin [5]. Histones are highly specialized proteins making macromolecular complexes with the DNA that are referred to as nucleosomes [6].

Variants of the histone proteins are by far the most abundant in the H2A family, in which H2A1, H2A2, H2AX, and H2AZ have been described [7,8]. In normal human fibroblasts, two H2A molecules are present in each nucleosome and 10% of them are H2AX. Therefore, on average, there is an H2AX molecule every five nucleosomes [9].

In its C-terminal tail, at position 139, H2AX contains a single serine that is highly conserved from plants to humans, suggesting a crucial role throughout evolution [10]. In their seminal 1998 paper, Dr. Bonner’s group reported that γ phosphorylation of H2AX at serine 139 occurred a few minutes after DNA damage induced by ionizing irradiation [2]. The first demonstration for the role of H2AX in the DNA damage response (DDR) has come from a study in *S. cerevisiae*, where it was shown that elimination of the unique C-terminal H2A serine residue led to an impairment in non-homologous end joining (NHEJ), one of the two evolutionary conserved pathways for the repair of DNA DSBs [11]. Analysis of H2AX-deficient embryonic stem cells and mice has then shown that, in mammalian cells, H2AX is not essential for NHEJ or homologous recombination (HR), the other main pathways of DSB repair, but modulates both [12,13,14,15]. Today, we know that γH2AX serves as a docking cushion for the accumulation and retention of the components of the DDR [16]. Following the percentage of representation among H2AX variants in the chromatin, it was also estimated that a single DNA DSB causes H2AX phosphorylation to spread over up to two Mbp regions of chromatin, comprising nearly 2000 γH2AX molecules [2,17].

γH2AX is deemed as a crucial player in DDR because it is capable of inducing signals for both the DNA damage-sensitive cell cycle checkpoints and the DNA repair proteins [18] (Figure 1). The association between γH2AX and the mediator of DNA damage checkpoint protein 1 (MDC1) is one of the first phases during which the DSB is arranged for DNA damage signaling and repair [19]. To this initial step, several other signaling and repair proteins, such as the p53 binding protein 1 (53BP1) and the breast cancer gene protein 1 (BRCA1), accumulate at DSBs with the intervention of γH2AX [20]. Therefore, γH2AX is a sensor, present in the early recognition checkpoints of DNA damage. Once H2AX has been phosphorylated, the MRN complex forms, made of three DSB repair proteins: the MRE11 homolog DSB repair nuclease (MRE11), the RAD50 DSB repair protein (RAD50), and the Nijmegen breakage syndrome 1protein (NBS1) also referred to as nibrin [19]. The MRN complex plays a pivotal role in the response to DNA damage and maintenance of chromosomal integrity [21,22]. The signals initiated by γH2AX and other sensors are transmitted, among others, to the p53 tumor suppressor protein and the cdc25 family of phosphatases [18]. Then, a series of transducer proteins follow, among which the ataxia telangiectasia mutated (ATM) kinase. The mechanism by which DNA damage activates ATM relies on cyclin-dependent kinase 5 (Cdk5) that, once activated by DNA damage, directly phosphorylates ATM at Ser 794 [23]. Phosphorylation at Ser 794 precedes and is required for ATM autophosphorylation at Ser 1981, the last step to initiate ATM kinase activity. The Cdk5-ATM signal regulates the phosphorylation and function of the ATM targets cellular tumor antigen p53 (p53) and H2AX. Interruption of the Cdk5-ATM pathway attenuates the DNA damage-induced neuronal cell cycle re-entry and the expression of the pro-apoptotic p53 upregulated modulator of apoptosis (PUMA) and apoptosis regulator Bcl-2-associated X protein (Bax), protecting neurons from death. Thus, activation of Cdk5 by DNA damage serves as a critical signal to initiate the ATM response and regulate ATM-dependent cellular processes. The serine/threonine-protein kinase ATR can also phosphorylate H2AX, chiefly in response to DNA replication stress. Lastly, γH2AX accumulation attracts repair factors, increasing the concentration of repair proteins surrounding a DSB site and insures DDR [24,25,26,27].

### 1.2. H2AX, Cell Cycle Checkpoints, and Genomic Instability

Cell cycle checkpoints are the surveillance machinery that controls the sequence, integrity, and fidelity of the chief events of the cell cycle [28]. These events comprise the regulation of cell size, the replication and integrity of the chromosomes, and their correct segregation at mitosis. Many of these mechanisms are antique in origin and highly conserved in evolution while others have evolved more recently in higher organisms, and control unconventional cell fates with notable impact on tumor suppression.

H2AX has an important role in the G2/M checkpoint (Figure 1), one of the checkpoints controlling cell size and its coordination with cell cycle progression [29,30]. Although rapidly phosphorylated in response to ionizing radiations, H2AX is largely superfluous for checkpoint responses to high-dose γ-irradiation, as the latter causes immediate death of the cell [11,13,31]. On the other hand, cells lacking H2AX that are exposed to low doses of irradiation generating few DSBs do not properly arrest at the G2 phase of the cell cycle and move to mitosis [32]. This malfunctioning G2/M checkpoint response seems to be linked to the decreased accumulation at DSBs of 53BP1, which may be essential to increase the DDR at threshold levels of DNA damage [13,33,34].

The result of inaccurate DNA repair or deficiency in cell cycle checkpoints is a predisposition for an increased chromosomal pathology, which is commonly referred to as genomic instability [35]. Classically, the instability shows as chromosomal breaks, translocations, or aneuploidy.

### 1.3. H2AX and Cell Growth

H2AX’s involvement in the control of cell growth appeared to be evident after observing that H2AX^−/−^ mice are smaller than their normal littermates [13]. Likewise, H2AX^−/−^ mouse embryonic fibroblasts display reduced growth and senesce after only a few passages in culture [13]. Telomeres, the protecting ends of linear chromosomes, become shorter at every cell division, and their shortening is proposed to be a primary cause of aging, cellular senescence, and numerous diseases [36]. γH2AX has been detected in fusogenic telomeres of cells depleted of the capping protein telomeric repeat-binding factor 2 (TRF2), as well as in fibroblasts undergoing senescence [37,38]. However, standard mitotic telomere preservation does not require H2AX, and the histone is unnecessary for the chromosome fusions arising from either critically shortened or deprotected telomeres [39]. Therefore, the growth defect in H2AX-null mice is likely a consequence of the response to chromosomal aberrations arising spontaneously in primary cells, rather than the result of structural modifications in telomeres [3].

### 1.4. H2AX and Mitosis

Post-translational modifications of histones intervene in the structural and functional regulation of chromatin during cell division. In 2005, it was documented that, in the absence of exogenously or artificially induced DNA damage, H2AX was highly phosphorylated in normally growing mammalian cells [40]. In this report, the authors described the existence of two distinct patterns of γH2AX immunostaining. Large amorphous foci, morphologically comparable to the foci appearing in response to irradiation-induced DSBs, were detected in a quite small population of cells. In these foci, several DDR proteins were also identified and, thus, this pattern of staining was interpreted as revealing the natural occurrence of DSBs. In addition, another much more plentiful population of small foci was observed that did not stain for the main proteins involved in DNA DSB repair. The γH2AX signal intensities increased as cells moved from G1 into S, G2, and M phase of the cell cycle, with peaks at metaphase. The presence of γH2AX during mitosis was thereafter confirmed by other groups [41,42,43]. Several hypotheses were then put forward based on the idea that H2AX contributed somewhat to the fidelity of cell division, and it was suggested that phosphorylation of H2AX might also affect chromatin condensation and transcriptional inactivation during the repair of DNA DSBs in mitotic cells. More recent work has demonstrated the intervention of the histone in the formation of an intact mitotic (S/M) checkpoint complex [44,45].

Remarkably, it was also demonstrated that Aurora B kinase, a protein that functions in the attachment of the mitotic spindle to the centromere, phosphorylated histone H2AX, but at serine 121 (H2AX-pS121), and thus promoted its autophosphorylation, this mechanism being essential for proper chromosome segregation [46].

### 1.5. H2AX and Embryogenesis

After zygote formation, substantial changes in gene expression follow a genome-wide chromatin remodeling [47]. H2AX is among the histone variants highly expressed in undifferentiated mouse embryonic stem cells immediately after fertilization and preimplantation embryos [48,49], and γH2AX was highly expressed throughout preimplantation development in the absence of experimentally induced DNA damage [50].

Other work has hypothesized the intervention of γH2AX in the maintenance of the differentiation state of mouse embryonic stem cells. High levels of γH2AX have been related to global chromatin condensation in these cells rather than pre-existing DNA damage, with the occurrence of large foci of immunoreactivity that were not associated with, e.g., 53BP1 [51]. The γH2AX epigenetic modification was also demonstrated to support the self-renewal and proliferation capability of mouse embryonic stem cells [52]. More recently, it was observed that the regions of the embryonic stem cell chromatin in which γH2AX was specifically recruited correlated with certain silenced extraembryonic genes, and that H2AX deficiency led to upregulation of some extraembryonic genes but not of pluripotency genes or germ layer markers [53].

The role of H2AX in embryogenesis was also supported by studies in species other than the mouse [54,55,56].

Finally, it should be recalled here that embryonic stem cells divide very rapidly, and it is thus conceivable that H2AX activation is also linked to high replication stress, as embryonic stem cells are often in the S phase of the cell cycle with the occurrence of high levels of DNA single-strand breaks (SSBs) [57].

### 1.6. H2AX and Naturally Occurring DSBs in Gametes and Immune Cells

#### 1.6.1. H2AX and Gametogenesis

Gametogenesis requires a particular type of cell division referred to as meiosis, during which physiological DSBs are created and repaired, resulting in chromosome recombination, a process in which H2AX also participates [58]. Much of the research work on the intervention of H2AX in meiosis has been carried out in spermatocytes. Meiotic recombination occurs during the prophase of the first meiotic division and is prompted by the DSBs produced by the Spo11 transesterase [59]. The distribution of γH2AX in mouse spermatocytes showed two distinct patterns of staining in the autosomes and sex chromosomes [60]. The occurrence of γH2AX in sex chromosomes has a physiological role, as the X- and Y-chromosomes of histone H2AX-deficient spermatocytes did not form a sex-body (a macrochromatin body where the X- and Y-linked genes are transcriptionally repressed), did not start meiotic sex chromosome inactivation and displayed severe defects in the pairing of X-Y chromosomes [61].

Studies in human oocytes have disclosed some female-specific features of recombinational double-stranded DNA repair concerning synapsis and telomere dynamics [62]. Specifically, γH2AX patches were commonly present in female synaptic chromosomes, an observation that was rare in the male and suggested that synapsis installs faster than the progression of recombinational DSBs repair or that the latter is slower in the female [62]. Other observations in mammalian oocytes devoid of the synaptonemal complex protein 3 (SYCP3) have shown a strong residual γH2AX labeling retained at late meiotic stages in mutant oocytes and increased persistence of recombination-related proteins associated with meiotic chromosomes.

#### 1.6.2. H2AX and Lymphocyte Development

During lymphocyte development, T and B cells undergo a process of somatic gene rearrangement referred to as variable diversity joining (V(D)J) recombination to produce the primary antigen receptor repertoire [63]. The diversification of antigen receptors in lymphocytes requires the introduction of DSBs adjacent to the V, D, and J segments that form the antigen receptor [64]. As γH2AX foci accumulate at sites of V(D)J recombination in developing thymocytes [21], and a subgroup of T and B cell lymphomas in H2AX^−/−^p53^−/−^-deficient mice displayed oncogenic translocations arising as byproducts of aberrant V(D)J recombination, a role for H2AX in the process was hypothesized [65,66]. B lymphocytes undergo a second genomic recombination reaction in response to antigenic stimulation called class switch recombination (CSR), a process in which a single-stranded DNA deaminase makes lesions that are resolved by NHEJ, and is facilitated by γH2AX [15]. During CSR, numerous DSBs are created in the DNA of the Ig switch region. These DSBs can be resolved by local re-ligation or by recombination between switch regions. The latter is abnormal when H2AX is lacking, whereas short-range intra-switch region recombination proceeds normally [15].

### 1.7. H2AX and Mitochondrial Homeostasis

Mice lacking H2AX not only display limited growth (see Section 1.3) but also neurobehavioral alterations that were recently correlated to the impairment of mitochondrial function and repression of the mitochondrial biogenesis gene peroxisome proliferator-activated receptor γ coactivator 1α (PGC-1α) [67]. In this study, H2AX loss led to reduced levels of the major subunits of the mitochondrial respiratory complexes in mouse embryonic fibroblasts and the striatum, a brain region particularly vulnerable to mitochondrial damage. These defects were validated by the observation of a disrupted mitochondrial morphology in H2AX mutant cells and that the ectopic expression of PGC-1α restored mitochondrial oxidative phosphorylation complexes and mitigated cell death. It seems reasonable that, if investigated, a role of H2AX in mitochondrial homeostasis will be also found in other tissues.

### 1.8. H2AX and Apoptosis

Apoptosis, perhaps the most common form of programmed cell death (PCD) [68], is a physiological process essential for embryogenesis, morphogenesis, and tissue homeostasis, leading to a balance between cell survival and demise. In the nervous system, PCD is of paramount importance during the differentiation, maturation, and maintenance of the neurons and glial cells in the normal brain [69]. Notably, its deregulation is often causative of neurodegenerative diseases [70] and dementia [71].

A little while after the discovery of the importance of γH2AX in the DDR to DSBs, it was shown that H2AX γ phosphorylation also occurred during apoptosis [72]. Subsequently, it was demonstrated that the γH2AX staining pattern in apoptosis was different from the focal distribution observed in the DDR [73,74] (Figure 2).

Specifically, three different forms of staining were observed in temporal sequence: a nuclear ring staining in early apoptotic cells in the absence of clear modifications of the size of the nucleus; then diffuse nuclear staining, again without obvious nuclear alterations; and, finally, homogeneous staining of the apoptotic cell bodies [75]. Studies on the apoptotic ring have mainly been focused on tumor cell lines or primary cells (e.g., prostate epithelial cells, blood white cells) after an experimental challenge to induce apoptosis [75]. After immunocytochemical detection, the ring was demonstrated to contain several molecules besides γH2AX, such as phosphorylated ATM on Ser1981 (P-S1981-ATM), phosphorylated checkpoint kinase 2 on Thr68 (P-T68-Chk2), phosphorylated DNA-dependent protein kinase on Thr2609 (P-T2609-DNA-PK), phosphorylated histone H2B on Ser14 (P-S14-H2B), and phosphorylated heat shock protein 90 on Thr5/7 (P-T5/7-HSP90), and to colocalize with staining for cCASP3 [75]. It is important to remark here that after induction of a DDR by, e.g., ionizing radiations, γH2AX appears in foci, and only thereafter do cells initiate apoptosis and form apoptotic rings, whereas when apoptosis is primarily induced by pro-apoptotic agents that do not damage the DNA, the pattern of γH2AX immunostaining directly appears in the form of the apoptotic ring (see Figure 4 in [75]). There are also molecular differences between the apoptotic and DDR γH2AX signals: the DDR foci contain MDC1 and 53BP1 but not P-S14-H2B, whereas the apoptotic γH2AX signal contains pS14-H2B but neither MDC1 nor 53BP1 [73].

### 1.9. H2AX and Hereditary Syndromes

Some hereditary diseases disturb the cellular response to DSBs. These diseases can be collectively referred to as chromosomal instability syndromes as they are associated with chromosomal instability and breakage [76]. Among them, there are three autosomal recessive disorders: ataxia-telangiectasia, Nijmegen breakage syndrome, and Bloom’s syndrome [3], and Fanconi anemia that can be inherited as autosomal recessive, autosomal dominant, and X-linked [77].

Ataxia-telangiectasia is a disorder that primarily produces cerebellar ataxia. It results from a mutation in the ATM gene with total or partial loss of the ATM protein [78]. Nijmegen breakage syndrome is a result of mutations in the nibrin gene and is associated with immunodeficiency [79]. Bloom syndrome is caused by a lack of the Bloom syndrome protein (BLM) helicase [80]. Fanconi anemia is a DNA repair disorder that eventually leads to chromosomal instability, particularly upon exposure to cytotoxic therapies, and a predisposition to certain tumors [77].

H2AX has been implicated in all these four pathologies [81,82,83,84] that are primarily characterized by growth defects, immunodeficiency, hypogonadism, hypersensitivity to specific DNA-damaging agents, chromosomal fragility, and cancer susceptibility. Remarkably, the mouse models carrying targeted mutations in genes affecting the biological responses to DNA damage (see Section 1.1 above) sum up most of the pleiotropic features of chromosomal instability syndromes [85].

### 1.10. H2AX and Tumors

Staining for γH2AX has been proposed as a marker of DNA damage and genomic instability in cancerous cells [86]. This is also because increasing evidence has accumulated that genomic instability may be causative and not only consequential of cancer development [87]. The DDR is typically triggered in response to oncogenes and throughout the early stages of solid tumor growth. Thus, the detection of γH2AX can theoretically serve as a biomarker for the transformation of normal cells to the premalignant and malignant stages. γH2AX has been studied in many cancer types, among which breast, lung, colon, cervix, and ovary cancers [88]. In the case of the brain, H2AX phosphorylation has been described in glioblastoma, pediatric medulloblastoma, and neuroblastoma [88]. Translationally, it is remarkable that there is little or no increase in carcinogenesis in H2AX^−/−^ mice, although chromosomes in cells from these mice have frequent breaks and translocations [65]. It seems that p53 protects H2AX^−/−^ cells from malignant transformation. There is a remarkable increase in the onset of different types of tumors when the loss of one or both H2AX alleles occurs in p53-deficient mice [65,66]. These experiments demonstrated that H2AX was still expressed in tumors from H2AX^+/−^p53^−/−^-deficient mice, implying that H2AX works as a dosage-dependent or haploinsufficient tumor suppressor [89].

## 2. H2AX in the Normal Brain

### 2.1. H2AX in the Developing Brain

The development of the central (CNS) and peripheral nervous systems requires a regulated balance between the generation of new cells (neurons and glia) and their apoptotic demise. Remarkably, evidence is accumulating that H2AX has a role in both these processes (Figure 3). In support, a study on embryonic kidney cells has demonstrated that EYA, a protein tyrosine phosphatase, is implicated in promoting effective DNA repair rather than death in response to genotoxic stress by dephosphorylating an H2AX carboxy-terminal tyrosine phosphate (Y142) in a damage signal-dependent manner. This post-translational modification determines the relative recruitment of DNA repair or pro-apoptotic factors to the tail of γH2AX, allowing it to function as an active determinant of repair/survival versus apoptotic responses to DNA damage [90].

Strictly speaking, neurogenesis is defined as the process of generating new neurons. In its broader definition, the term also comprises the production of new glia, which can be more specifically defined as gliogenesis. In the following, we will simply use the term neurogenesis to also comprise gliogenesis.

Data have gathered in the past decades demonstrating that neurogenesis not only occurs during embryonic development but proceeds all along with the first weeks/months postnatally, until, at least in certain brain areas, adulthood and old age (Figure 3). We will here consider adult neurogenesis, often improperly referred to as postnatal neurogenesis, in Section 2.2.

Embryonic neurogenesis is a well-understood process in vertebrates and, specifically, all eutherian species; remarkably it proceeds for some time after birth in certain areas of CNS in altricial animals, i.e., those species where newborn subjects require long parental care, including humans [91]. This neurogenetic temporal window can be more precisely referred to as infantile neurogenesis, as the term postnatal neurogenesis has become somewhat unclear after the recognition of adult neurogenesis, which also occurs postnatally.

#### 2.1.1. γH2AX in Embryonic Neurogenesis

The billions of cells of the mature brain are generated during embryonic life from stem cells and neural progenitors located in all regions of the neural tube. According to recently reviewed terminology [92], stem cells are cells that can undergo self-renewing divisions producing additional stem cells with the same properties and potential or divisions that generate daughter cells differentiating into multiple cell types. Neural progenitor cells, instead, are cells producing many, if not all, of the glial and neuronal cell types of CNS. However, they do not produce non-neural CNS cells, such as the immune system cells, i.e., the microglia and T and B cells, and dendritic cells [93]. It is not surprising that H2AX may be activated during embryonic neurogenesis, as its phosphorylation also occurs during mitosis (see Section 1.4 above), which is required for the generation of neurons and glia from stem cells and/or neuronal progenitors.

Neurogenesis in the embryo is concomitant with the transformation—at E 14.5 in the mouse—of neuroepithelial stem cells into radial glial stem cells (RGSCs), which directly or indirectly—with the intermediate formation of a basal progenitor—give rise to the neurons [94] (Figure 3).

To the best of our knowledge, we have been the first to describe γH2AX immunoreactivity in the mouse embryonic CNS [95]. We detected high levels of γH2AX particularly in the subventricular zone (SVZ) of the lateral ventricles, rostral migratory stream (RMS), and the olfactory bulb (OB) system of E 14.5 embryos. Double immunofluorescence (IMF) labeling for γH2AX and the proliferation marker phosphorylated histone H3 (pHH3 [96]) at E 14.5 showed the existence of two different populations of γH2AX proliferating cells in the developing cerebral wall with M (labeling of condensed chromosomes) and G2 (focal staining) phase morphologies (Figure 3). Although we did not attempt any type of molecular characterization of these cells, their positions suggest that those in the ventricular zone (VZ) and SVZ were progenitor cells and proliferating RGSCs, whereas γH2AX immunoreactive cells in the intermediate zone (IZ) were basal progenitors (BPs), also referred to as intermediate progenitor cells (IPCs). In keeping with our observations, it was previously reported that a mouse pluripotent embryonic stem cell line (Nanog-ESCs) had high basal levels of γH2AX that were correlated to its self-renewal capacity [52]. In addition, GABA_A_ receptors were demonstrated to regulate neural stem cell proliferation during the development of the cerebral cortex by inhibiting DNA synthesis [97], a mechanism that was dependent on H2AX [98,99]. Our results indicate the existence of a DDR during the differentiation of the cerebral cortex. Such a DDR could be a consequence of the considerable onset of DSBs during the intense proliferative activity of neuroepithelial precursors, RGSCs, and BPs at the E 14.5 stage. In accord with this possibility, the loss of the breast cancer type 2 susceptibility protein (BRCA2), another protein involved in DSB repair and/or HR, results in defective embryonic (and postnatal) neurogenesis [100]. In addition, a recent cytofluorimetric study has demonstrated that actively replicating neural progenitor cells harbor more DNA DSBs during early neurogenesis than at later differentiation stages [101]. Moreover, a study on mice lacking recombination activating 2 (RAG-2), a component of the RAG-1,2-complex responsible for initiating somatic recombination in lymphocytes, led to the identification of the long interspersed element-1 (LINE-1) retrotransposition as one of the sources of DSBs in retinal neurogenesis and to the suggestion that both DSBs generation and repair are genuine processes intrinsic to neural development [102]. Our understanding of the role of γH2AX in embryonic neurogenesis is made more complex by the concomitant occurrence of apoptosis during the generation of new neurons and glia. Interestingly, in RAG-2-deficient mice, γH2AX immunoreactive foci were less abundant in the retina, retinal ganglion cell death was increased, and axonal growth and navigation were impaired. All these features were common with those in mutant mice defective in DNA repair mechanisms [102].

We will discuss in more detail the relationship with cell proliferation, apoptosis, and H2AX phosphorylation in Section 2.1.2 and Section 2.2.

#### 2.1.2. γH2AX in Infantile Neurogenesis

A few brain areas, including the cerebellum and the prefrontal cortex, continue adding new neurons in infancy. To these, the SVZ/RMS/OB system and hippocampus should be added, where neurogenesis is believed to continue throughout life (see Section 2.2.1). In humans, a limited window of neurogenesis (before 18 months) has been reported to occur in the SVZ/RMS to reach the prefrontal cortex [103]. This process can thus be considered as an early event in the adult neurogenesis taking place in the SVZ/RMS/OB system. When we have studied the expression of γH2AX in P0-20 mice, we have observed positive staining in the SVZ/RMS/OB all along with this time frame—with a peak at P10-15—and in the cerebellum—with a peak at P5-10, whereas positive cells were detected in the subgranular zone of the dentate gyrus (DG) of the hippocampus only starting at P10, with low intensity [95].

We will therefore limit our discussion of the intervention of γH2AX in infantile neurogenesis to the SVZ/RMS/OB and cerebellum. It should be, however, remembered that neurogenesis continues in the SVZ/RMS/OB without interruptions in adult and old age and that we here make a temporal distinction with infantile neurogenesis only for the sake of simplicity.

##### 2.1.2.1. SVZ/RMS/OB

The histological features of γH2AX immunoreactive cells in the SVZ/RMS/OB during the perinatal period were remarkably similar to those reported in the previous section, where we described the occurrence of the phosphorylated form of the histone at E 14.5.

We also examined the ultrastructure of the SVZ in P6 mice that survived for two hours after a single 5-Bromo-2′-deoxyuridine (BrdU) injection using double immunogold staining procedures at the transmission electron microscope (TEM) level [95]. Based on their ultrastructural features, most γH2AX immunoreactive cells in SVZ could be identified as type B (astrocytes, i.e., the RGSCs giving rise to the neuroblasts) or type C cells (transit-amplifying cells). Some of them were also labeled for BrdU (see Figure 4A,B in [95]); in addition, some type A cells (migrating neuroblasts) were also double-stained (see Figure 3 for a graphical representation of the different cell types in the SVZ). Type B and C cells are proliferating elements and thus their incorporation of BrdU was not surprising. Additionally, it is of relevance that we did not observe cells with the classic apoptotic features after TEM examination, in keeping with the suggestion that most of the dividing cells in the SVZ of young rats did not undergo apoptosis [104], although apoptotic cells were detected in the SVZ of two-month-old mice after single staining for activated (cleaved) caspase 3 (cCASP3) [105]. Therefore, one cannot exclude that at least part of the γH2AX immunostaining in the first two months of postnatal life in rodents is related to apoptosis.

##### 2.1.2.2. Cerebellum

In altricial animals, i.e., the animals with inept offspring, much cerebellar development occurs postnatally, within a temporal frame of about two to three weeks in rodents and rabbits and up to three months in humans. At these ages, the cerebellar cortex is made of four layers (Figure 3) [106]. The external granular layer (EGL) is highly proliferative and made of pre-migrating neuroblasts, the precursors of the mature cerebellar granule cells (CGCs); at the same time, these highly proliferating cells undergo massive apoptosis before starting to migrate to the internal granular layer (IGL) [107]. After studying the cerebellum of P0-20 mice, we observed intense staining for γH2AX in the EGL, where immunoreactive cells reached a peak at P5-10 and started to decline from P15 [95]. Some γH2AX-positive cells were also seen in the IGL. After γH2AX+pHH3 IMF, double-labeled nuclei displayed M and G_2_ phase morphologies. γH2AX-IR apoptotic GCCs showed specific staining of their nuclei and were at different stages of the apoptotic process. Our observations thus suggest that also in the cerebellar cortex, there is an intense DDR as a consequence of the formation of high levels of DSBs during the intense proliferative activity of CGCs precursors [108]. This suggestion is in line with the observation that loss of BRCA2 resulted in anomalous cerebellar neurogenesis in a p53-dependant manner [100] and that an ATM-mediated DDR took place in cerebellar organotypic cultures [109]. It should be noted that the combined loss of *Nbs1* and *Atm* abrogated the DSB response with growth retardation, severe ataxia, microcephaly, cerebellar developmental defects, and appearance of γH2AX and 53BP1 immunoreactive nuclear foci in animals that did not survive beyond three weeks [110]. Remarkably, *Nbs1* knock-out mice displayed a phenotype that closely resembles that of human Nijmegen breakage syndrome (see Section 1.7), which is characterized by a dramatic decline in the number of CGCs and Purkinje neurons [111,112].

In the mouse, the Purkinje neurons are generated between E10 and E13 in the ventricular zone of the primitive neural tube [113]. After migration, these cells reach the cerebellar cortex within the first week after birth and undergo a CASP3-dependent apoptotic plateau [114,115,116].

It seems therefore reasonable to conclude that phosphorylation of H2AX in the postnatal cerebellum is strictly related to the proliferative (CGCs) and apoptotic (CGCs and Purkinje neurons) events occurring at this developmental stage. The well-documented intervention of CASP3 in the execution of cerebellar apoptosis (see [107]) is also in accord with our very recent observations on the link between γH2AX and cCASP3 in the old mouse brain [117] (see Section 2.2).

### 2.2. γH2AX in the Adult Brain

#### 2.2.1. γH2AX in Adult Neurogenesis

##### Classical Neurogenetic Areas

The two brain regions where adult neurogenesis has been best investigated are the hippocampus and the SVZ/RMS/OB system (Figure 3). In both regions, neurogenesis can be subdivided into well-defined stages, including cell proliferation, neuronal differentiation and growth, and functional integration into brain circuits (see, e.g., [118]). The hippocampal and SVZ/RMS/OB neurogenesis in the adult is, as a matter of fact, the continuum of the earlier process that begins during embryonic life and persists in infancy (see Section 2.1.1 and Section 2.1.2.1 above). Studies on neural stem cells during adult neurogenesis in the SVZ demonstrated that H2AX continued to play a role in the regulation of the process in adult animals as well [98]. Such regulation was dependent on GABA_A_ receptors to negatively control stem cell proliferation in vitro [98,99].

We have demonstrated the occurrence of γH2AX in both the SVZ/RMS/OB system and hippocampus of adult mice after semiquantitative estimation of the number of positive nuclei in IMF procedures [95]. In the former, γH2AX persisted at medium-to-high levels during the entire course of adulthood, whereas in the latter, the level of immunoreactivity was lower. In both areas of the forebrain, the number of immunoreactive nuclei was higher in younger animals and progressively declined with age. Such an observation was consistent with the result obtained after induction of a DDR with ionizing radiation exposure in mice, where younger animals were much more susceptible [119].

##### Others

Expression of γH2AX was detected using immunocytochemistry in several other areas of the adult brain (P20–P60) including the dorsal endopiriform nucleus, the midbrain, and pons [95]. Interestingly, the dorsal endopiriform nucleus, a derivative of the lateral pallium [120], has recently been demonstrated to contain developing brain homeobox 1 (*Dbx1*)-expressing neuronal progenitors [121], and thus it may be speculated that the expression of γH2AX in this nucleus is linked to regulation of these progenitors. The midbrain has also been proposed (with many controversies) as an additional site of adult neurogenesis (see, e.g., [122]), as well as the pons; in this latter case, however, chiefly after lesion of the vestibular system (for a recent review, see [123]).

#### 2.2.2. Cerebral Cortex

Although there are some claims that adult neurogenesis also occurs in the cerebral cortex of certain mammalian species (see, e.g., [118]), cortical neurons are largely acknowledged to be postmitotic. Therefore, the detection of γH2AX in the cerebral cortex of normal adult mice [95] somehow opens a new line of discussion.

Detection of γH2AX in the mouse cortex was stated to be evident already at P15 (thus in the mid-to-late stage of infancy) and occurred throughout adulthood to eventually continue in the old age (see Section 2.3). γH2AX immunoreactivity was mainly observed in nuclei belonging to NeuN-positive neurons, but also some GFAP immunoreactive astrocytes had a γH2AX positive nucleus. There was no co-localization with the proliferation marker pHH3 [95].

In the following, we will discuss the literature on the possible significance of γH2AX in postmitotic cortical neurons, with special attention to the (possible) relationship of γH2AX and cell proliferation. Much of the work carried out on the matter has been focused on the γH2AX response to different types of experimental stimuli rather than on analyzing the normal brain. Very interestingly, one of the first papers in the field reported that slightly cytotoxic stimulation of ionotropic glutamate receptors by NMDA was sufficient to evoke γH2AX in cultured cortical neurons and that some γH2AX foci co-localized with MRE11, one of the proteins of the DDR. These results indicated that at least a portion of γH2AX foci was damage dependent and corresponded to the extent of γH2AX observed following irradiation at a 1 Gray dose [124]. Another paper has investigated the role of the checkpoint kinase 1 (Chk1) on the viability of cultured rat cortical neurons that were not subjected to any type of DNA damage and reported that the specific inhibition of this kinase caused apoptosis of differentiated neurons with the appearance of DSBs, γH2AX, and p53 [125].

Numerous reports have investigated the γH2AX response to ionizing radiations. We will address these studies in Section 3.1, but it seems worth mentioning here a paper in which, using transmission electron microscopy to localize pKu70 and pDNA-PKcs, the formation and repair of DSBs within euchromatin and heterochromatin were monitored in cortical neurons [126]. It was thus observed that DNA lesions in euchromatin were labeled for pKu70 and quickly sensed and rejoined, whereas, in heterochromatin, DNA packaging retarded the processing of DSBs with the appearance of complex pKu70-clusters. All pKu70-clusters disappeared within 72 h post-irradiation, indicating efficient DSB rejoining. However, in heterochromatin, there were persistent 53BP1 clusters, occasionally co-localizing with γH2AX but not pKu70 or pDNA-PKcs, which possibly reflected an incomplete or inaccurate restoration of the chromatin structure rather than persistently unrepaired DNA damage [126].

Finally, several studies in different neuropathologies have investigated the effects of early DNA DSBs accumulation in cortical neurons and glia using γH2AX immunocytochemistry (see also Section 3.1.6). In Alzheimer‘s disease (AD), studies on autopsy material have shown an early neuronal accumulation of DNA DSBs [127]. Remarkably, it was also demonstrated that damage to brain endothelial cells could be or not independent of the AD pathology, in parallel with that in neurons and glia [128]. On the other hand, another human post-mortem study on amyotrophic lateral sclerosis (ALS) did not find a correlation between the expression of p16 and p21, two senescence markers, and γH2AH in the motor cortex [129].

Collectively, all the studies mentioned so far converge to suggest that expression of γH2AX in the adult cerebral cortex is related to some sort of DNA damage that could derive from a series of heterogeneous factors undermining the genomic integrity of the brain cells.

### 2.3. γH2AX in the Aging Brain

We have recently focused our attention on the expression of γH2AX and its relationship with apoptosis in the old mouse brain [117]. To better help readers with the following discussion, we have summarized our most relevant data in Figure 4.

#### 2.3.1. Neurogenic Areas

In mice older than 24 months, γH2AX immunoreactive nuclei were detected in the SVZ/RMS/OB system [95,117] and the hippocampal DG [117]. These two areas of the forebrain retain some neurogenic capacity also in the old brain and the former, in particular, was positively stained already during embryonic life [95] (Figure 3).

##### SVZ/RMS/OB

The SVZ/RMS/OB system is formed by three anatomically and functionally connected structures: the SVZ where neuroblasts are generated to migrate along the RMS and to provide new inhibitory granule cells and glomerular cells in the OB [130]. We demonstrated a statistically significant difference in the percentage γH2AX+BrdU double-labeled cells in the SVZ/RMS (the proliferative compartment) compared to the OB (the receptive compartment), with a reduction to 0.41-fold in the latter [95]. These observations were fully in line with the notion that the migration of neural precursors from SVZ to the OB along the RMS still occurs in the old brain [105,131,132]. More recently, we have demonstrated that the reduction of γH2AX in the foremost part of the system was consequent to the occurrence of DDR mechanisms in proliferating neuroblasts of the SVZ and that some of these γH2AX immunoreactive cells died during their journey to the OB. These observations are in line with the demonstration of persistent DNA damage in senescent cells and aged mammalian tissues [133] and with previous observations on the occurrence of higher numbers of apoptotic cells in the old mouse SVZ [105].

The authors also reported a reduction of BrdU incorporation in the SVZ of elderly mice, and it was speculated that increased cell death was responsible for pulling down the number of S phase cells in old animals [105], in parallel with major cytoarchitectural (atrophy) and proliferative changes inside the SVZ/RMS/OB neurogenic niche [134]. It is also interesting to note that telomere erosion was demonstrated to dramatically impair the in vitro and in vivo proliferation capacity of the SVZ adult neural stem cells in telomerase-deficient adult mice [135] and that stem cell numbers do decline in the aging mouse SVZ [136].

In particular, in response to aging and the consequent modifications of the niche microenvironment, a variety of aberrations have been described in the SVZ, including reduced cell proliferation, diminished neuroblast number, RMS thinning, and decreased expression of stage-specific developmental markers [105,137,138]. As recently reviewed [139], regular cell cycle machinery has been assumed to be of paramount importance for normal aging in the SVZ/RMS/OB system. Among the observed alterations of this machinery is an elongation of the cell cycle [140], resulting from a specific lengthening of G_1_ [141], cellular senescence [137], decreased telomerase activity [135], and enhanced type B cell (RGSC) inactivity [131]. As a consequence of these changes, the SVZ neurogenic potential abruptly declines by 50–75% in the old brain [136,142,143] notwithstanding the uninterrupted generation of few newborn neurons [137,138]. Finally, an increase in SVZ cell death, typically in neuroblasts, was also detected in old- but not middle-aged mice [105]. Interestingly, apoptosis can produce a feed-forward degenerative cell cycle in the old brain leading to neurodegeneration with an accumulation of unrepairable DNA DSBs and acquisition of a senescent cellular state [144,145,146,147].

##### Hippocampus

Much likely, γH2AX-immunoreactive cells in the hippocampus DG correspond to IPCs and neuroblasts (Figure 3). The hippocampus to some extent displayed a different pattern of γH2AX immunostaining when compared to the SVZ/RMS/OB system, specifically regarding the colocalization of the phosphorylated histone with DNA synthesis/cell proliferation markers [95]. Therefore, the question was left open of whether activation of H2AX in the DG was related to the occurrence of DNA DSBs during cell division, apoptosis, or simply to physiological activity [148]. In the old mouse hippocampus, similarly to the SVZ/RMS/OB, the expression of γH2AX was accompanied by 53BP1, and cCASP3, and, to a small extent, BrdU immunostaining [117]. It is worth noting that the small cell population of the hippocampus that incorporated BrdU in part also expressed pHH3 [117], making it more difficult to correctly deduce the function of γH2AX in hippocampal cells. Remarkably, an increase in DSBs (measured with γH2AX immunocytochemistry) was observed in the six-month-old mouse hippocampi, where neuronal activity was experimentally amplified and exacerbated by amyloid β, with an aggravation of synaptic dysfunctions [148]. Thus, it seems reasonable to suppose that aged hippocampal cells expressing the DDR markers γH2AX and 53BP1, as well as activated CASP3, have accumulated an amount of DSBs that was incompatible with survival. So, the incorporation of BrdU in these cells might have been linked to an aberrant re-entry into the cell cycle before undergoing apoptosis [147] (Figure 4). The wild-type p53-induced phosphatase 1 (Wip1) critically regulates DDR under stressful situations [149], and Wip1 gene knockout (KO) mice showed aberrant elevation of hippocampal cellular senescence and expression of γH2AX [150]. Thus, the lack of Wip1-mediated γH2AX dephosphorylation facilitates cellular senescence in this area of the forebrain.

Remarkably, a study on autopsy hippocampal tissue samples from patients with AD neuropathology has demonstrated that high levels of oxidative DNA damage (detected using 8-Hydroxyguanine and γH2AX) were present in all groups of subjects independently from Braak classification [151]. However, in subjects with neuropathological AD and clinical dementia, there was a reduction in DNA repair with an increase in cell cycle progression to death when compared to those with neuropathological AD without cognitive impairment and normally aging individuals. Moreover, there were no differences in biomarker expression between patients with neuropathological AD without cognitive impairment and normally aging individuals. The authors reported both nuclear and cytoplasmic γH2AX staining of the hippocampal neurons. The last observation is of interest and prompts some debate as neurons and glia in mild cognitive impairment (MCI) and AD patients, as well as in mice where neuronal activity was increased with kainate, showed a diffuse pan-nuclear pattern of γH2AX immunoreactivity that was different from the focal expression of γH2AX after X-ray irradiation [127].

Thus, the functional significance of the expression of γH2AX in the normal aging hippocampus remains still to be fully understood as from one side, it appears that neuronal activity can increase (pan nuclear?) γH2AX, whereas normal and pathological aging appear to be related to the occurrence of some form of DNA damage and the consequent activation of a DDR with the onset of the classic focal nuclear expression of the histone in phosphorylated form.

#### 2.3.2. Cerebral Cortex

The cerebral cortex was one of the main sites of expression of γH2AX in old mice and about 50% of γH2AX immunoreactive cells were also stained for 53BP1 [95,117]. In these animals, the exclusively focal appearance of γH2AX (and 53BP1) immunoreactivity and the persistingly high percentage of γH2AX+BrdU colocalization (about 60%) strengthened the notion that a DDR was occurring in cortical cells. The percentage of these cells expressing the two labels was remarkably constant along the rostrocaudal axis of the telencephalon [95], suggesting that the old cerebral cortex in its totality is prone to DNA damage. We also showed that there was no colocalization of BrdU and pHH3 in the presence of subpopulations of γH2AX immunoreactive cells that also expressed BrdU or cCASP3 [117]. Therefore, we concluded that BrdU incorporation in the old cerebral cortex was part of a naturally occurring DDR with phosphorylation of H2AX, recruitment of 53BPI, and activation of CASP3, eventually leading to death (Figure 4). The volumetric density of cCASP3 immunoreactive cells was higher than that of the cells that incorporated BrdU. Thus, at least a fraction of the cCASP3-positive cells likely undergo apoptosis very quickly, having accumulated too many unrepaired SSBs, as these can be fixed more rapidly than DSBs [152].

It also seems realistic that other cells, in which DSBs are formed, initially tried to repair their DNA (and hence incorporated BrdU), and, if failing, eventually suffered CASP3-dependent apoptosis (Figure 4). We cannot exclude that a pseudo-DDR, i.e., the appearance of γH2AX foci in the absence of detectable DNA breaks, also occurred, as demonstrated in other types of cells undergoing senescence in vitro [153]. However, such an event seems highly unlikely, as there was no increase of 53BP1 in γH2AX foci of senescent fibroblasts [153], whereas we have observed focal 53BP1 in cortical cells. We also believe that cortical cells, which incorporated BrdU but were negative to γH2AX, were committed to death, in agreement with the known intervention of γH2AX in the modulation of checkpoint responses (see Section 1.1 and [154]).

DNA repair systems are very important in the adult and old brain [155,156,157], and effective DNA repair is certainly needed in long-living postmitotic cells that have no or limited regeneration capabilities from precursors. Yet, γH2AX immunoreactive neurons in the old cerebral cortex may also be senescent-like neurons surviving with a persistently activated DDR [147,158]. These senescent-like cells may be resilient to apoptosis but predisposed to inflammation and neurological dysfunction [147]. Remarkably, the functional consequences of an age-dependent activation of DDR with the formation of senescent-like neurons may be very meaningful for neurodegeneration, cognitive decline, and dementia [159,160].

Studies on different types of experimental paradigms and transgenic mice are in line with the observations reported so far. For example, chronic exposure to alcohol leads to H2AX phosphorylation and decreases the level of 53BP1 in cultured primary cortical neurons, leading to a defective DDR [161] and there is a persistent accumulation of unrepaired DNA damage as measured after γH2AX immunostaining in rat cortical neurons after irradiation of the whole brain [117,162,163,164]. In addition, cell nuclei retain persistent γH2AX foci after exposure to ionizing reactions, allowing γH2AX to accumulate in telomeric DNA and senescent cells [164]. A study on a transgenic mouse line has also highlighted the role of γH2AX in an ATM signaling pathway alternative to NBS1/MRN, particularly in response to oxidative stress and aging [165] The ATM interactor (ATMIN) is an essential component of this pathway and ATMIN-deficient young mice displayed a cortical distribution of γH2AX immunostaining that was not significantly different from that of normal mice (see Section 2.2.2) and was accompanied by moderate numbers of cells expressing active phosphorylated ATM (pS1987-ATM), the ATM substrate phosphorylated structural maintenance of chromosomes protein 1 (pS957-SMC1), and the p53 protein [166]. ATM activation only slightly increased with age in ATMIN-deficient old mice and was complemented by a clear increase of γH2AX, whereas in control mice, there was a substantial increase of pS1987-ATM in the presence of a limited number of γH2AX-positive cells. These observations indicated that ATMIN contributes to ATM activation during aging and reinforce the notion that expression of γH2AX in old age is indeed related to various sorts of DNA damage.

## 3. γH2AX in the Experimentally Damaged Brain and Neuropathology

### 3.1. γH2AX and DNA Damaging Agents

Given the well-established role of H2AX in the DDR, it is not surprising that numerous surveys have been devoted to understanding the response of H2AX to DNA damage under an array of experimental conditions. We will try to summarize below the most relevant findings to appreciate the multiple roles of γH2AX in the brain under pathological conditions.

#### 3.1.1. Ionizing Radiations

Many studies have investigated the γH2AX response to ionizing radiations in the developing, mature, and old brain. These studies were carried out in vitro and in vivo with X-rays or radiomimetic drugs. We have already mentioned some of them in the previous sections when important in the context of the discussion of the data gathered in the normal brain.

Studies in vitro assessing the DNA damaging effects of X-rays or radiomimetic drugs have been carried out on several different types of neuronal primary cells or cell lines. Among the former, early (1998) experiments on rat cortical neurons and astrocytes showed that X-ray-induced DNA damage caused phosphorylation of H2AX and neuronal apoptosis, that the amount of damage and the degree of apoptosis induced were directly correlated, that slow repair of damage could play a role in the susceptibility of neurons to apoptosis, and that astrocytes were relatively resistant to death [167]. Subsequently, the same group demonstrated that when cells were irradiated with moderate X-ray doses (≤32 Gray), neuronal death was induced but could be blocked in vitro by a pan-caspase inhibitor, whereas death was not prone to caspase inhibition after severe DNA damage (up to 128 Gray) and could occur by non-apoptotic (necrotic) mechanisms [168].

As regarding the behavior of different types of neurons in response to ionizing radiations, it is interesting to note that the disappearance of γH2AX foci after irradiation was slower in human embryonic stem cells compared to progenitors and astrocytes, and this observation has led to the hypothesis that the DSB repair machinery may be more complex in immature nerve cells [169]. Additionally, the in vitro inhibition of glycogen synthase kinase 3β (GSK3β), a serine/threonine kinase that controls a variety of biological processes including cell proliferation and apoptosis [170], speeded the DSB repair efficiency in irradiated mouse hippocampal neurons, in parallel with the diminution of radiation-induced γH2AX foci and p53 accumulation [169,171].

Other neurons that were demonstrated to be damaged following X-ray irradiation in vitro are the retinal photoreceptors [172]. These cells responded to the ionizing radiations by activating a canonical DDR with the phosphorylation of H2AX. Remarkably, in this experimental context, the oncogene promoter staphylococcal nuclease and Tudor domain-containing 1 gene (*SND1*) [173] was demonstrated to activate ATM signaling launching DNA repair, this revealing SND1 as a novel regulatory factor in DDR. Other studies on photoreceptors were carried out ex vivo. To assess the impact of chromatin relocation on the localization of DNA damage, γH2AX, and terminal deoxynucleotidyl transferase dUTP nick end labeling (TUNEL) foci induced by the radiomimetic drug bleomycin were investigated in H3K4me3-immunolabeled rod nuclei from four-week-old animals (when rods are fully differentiated G_0_ neurons) [174]. H3K4me3 is an epigenetic modification to the DNA packaging protein HH3. It is a mark that indicates the trimethylation at the 4th lysine residue of HH3 and is often involved in the regulation of gene expression [175]. Preferential localization of γH2AX foci in euchromatin was detected in this study, together with foci clustering and in parallel with a decay of H3K4me3 signal at γH2AX foci. Therefore, localization of DNA damage signals in the differentiated rods indicated that euchromatin is prone to damage in these cells.

Among the most commonly used neural cell lines are those derived from neuroblastomas. When several lines of these cells were treated with the radiomimetic drug neocarzinostatin—an enediyne antitumor antibiotic that mediates strand breakage of the DNA, or with a 10 Gray dosage of X-rays, it was demonstrated that two distinctly altered cellular responses to DNA DSBs occurred, but both were associated with the onset of γH2AX [176].

It is also not surprising that glia is damaged by exposure to ionizing radiations. Phosphorylation of H2AX was observed in the response of cultured normal human astrocytes to 10 Gray X-irradiation [177]. In this study, the Western blotting analysis showed that astrocytes displayed a strong increase in the expression of NHEJ DNA repair enzymes within 15 min post-irradiation and increased expression of HR DNA repair enzymes after near two hours, to return to basal levels about 48 h later.

The results obtained in vivo and/or ex vivo were largely reproduced in vivo. Thus, γ irradiation of the developing mouse brain with a 2 Gray dose induced—within 24 h—massive apoptosis of immature neural precursors but not of mature neurons [178]. The two types of cells displayed comparable numbers of nuclear foci of phosphorylated H2AX, suggesting that differences in their radiosensitivity were not related to variations in the number of DNA DSBs induced by radiation. Both immature and mature neurons lost γH2AX within 24 h after irradiation, but precursors had slower kinetics of loss of γH2AX foci. This implied that the slowness of the DDR in neural precursors was related to their greater radiosensitivity. The same authors also found diffuse γH2AX staining of nuclei of cells at an early stage of apoptosis, whereas cells at later stages of apoptosis were unstained.

The spatiotemporal dynamics of γH2AX in the mouse brain after acute irradiation at different postnatal days were also analyzed with special reference to the DG of the hippocampus, where an extensive age-dependent induction of γH2AX foci regions one day after whole-body gamma irradiation with 5 Gray at postnatal day 3 (P3), P10, and P21 was observed 24 h post-irradiation [163]. Very interestingly, persistent DNA damage foci were demonstrated in the brain 120 days and 15 months thereafter, and these mice had shortened lifespan compared to the age-matched controls.

We very recently demonstrated that whole-brain X-ray irradiation at 10 Gray induced strong phosphorylation of H2AX with the appearance of nuclear foci of immunofluorescence for γH2AX and 53BP1 that revealed the occurrence of a DDR in irradiated cells after 15 or 30 min survival [117]. Our observations indicated that the cerebral cortex, hippocampus, and SVZ/RMS/OB had different vulnerabilities to X-ray-induced DNA damage.

#### 3.1.2. Neurotoxic Substances

Among the numerous neurotoxic substances that are capable of inducing DNA damage, some have also been specifically associated with the ability to phosphorylate H2AX. One of the first to be investigated was paraquat (1,1′-dimethyl-4,4′-bipyridinium dichloride), a widely used universal herbicide, which was known to accumulate in the cerebral cortex at the highest levels among herbicides after systemic treatment, to impair hippocampal neurogenesis [179], and to possibly induce Parkinson’s disease (PD) in humans [180,181] and animal models [182]. The toxic effects of paraquat were associated with DDR in a study on cortical neurons that were demonstrated to undergo apoptotic DNA fragmentation with a focal expression of γH2AX after a three-day treatment in vitro with a herbicide concentration of about 10 µM [183].

A list of other substances that were capable to induce γH2AX in neurons and/or glia in vitro and/or in vivo is reported in Table 1 below.

#### 3.1.3. Oxidative Stress

Numerous studies have examined the phosphorylation of H2AX following different types of oxidative stress, as summarized in Table 2.

#### 3.1.4. Telomere Dysfunction

Telomere dysfunction induced by adenovirus-mediated expression of TRF2 was demonstrated to trigger a DDR with the formation of γH2AX nuclear foci and to activate ATM in primary embryonic hippocampal neurons and astrocytes, as well as neuroblastoma cells [22]. In the latter, TRF2 induced activation of p53, p21, and an upregulation of the senescence marker βgalactosidase. By contrast, in neurons, TRF2 increased p21, but neither p53 nor βgalactosidase was induced. These observations are interesting as DNA and telomere damage appear to intervene in the pathogenesis of AD [22]. More recently, the formation of telomeric repeat-containing RNA (TERRA) foci in highly proliferating mouse cerebellar neuronal progenitors and medulloblastoma was reported; in this study, however, TERRA foci did not colocalize with γH2AX [202], leaving open the comprehension of the possible role of H2AX in telomere regulation.

#### 3.1.5. Injury

Among the wide array of experimental injury models of the nervous system, some have investigated the occurrence of DNA damage by using, among others, γH2AX immunocytochemistry. So far studies have regarded a model of spinal cord damage [203] and brain ischemic injury following the occlusion of the middle cerebral artery [204].

#### 3.1.6. Neurologic Disorders

##### Alzheimer’s Disease (AD)

It is now widely accepted that repair of oxidative DNA damage, cell-cycle dysregulation, and neuronal death may influence the clinical manifestation of AD [151]. An altered DDR is regarded as an early pathogenic event [127,205] and γH2AX was shown to be increased in peripheral blood lymphocytes of AD patients [206]. Moreover, as tau proteins become hyperphosphorylated in AD and their alterations lead to a loss of nucleic acid safeguarding functions with the accumulation of DNA and RNA oxidative damage, it is notable that γH2AX foci were observed in KO-Tau neurons after heat stress [207].

##### Huntington’s Disease (HD)

Neuronal DNA damage is one of the major features of neurodegeneration in HD. Several observations converged to demonstrate that nuclear foci of γH2AX appear in ST*Hdh^Q111/111^* cells—an HD knock-in striatal cell line that expresses mutant huntingtin at an endogenous level, and neurons of the R6/2 mouse, an HD model [208]. In addition, it was observed that imbalanced levels of non-phosphorylated and phosphorylated BRCA1 contributed to the DNA damage response in HD as a consequence of the interaction of BRCA1 with γH2AX domain, an association that was reduced in HD [208]. Other authors have described a novel manganese-dependent ATM-p53 signaling pathway that was selectively impaired in patient-derived neuroprogenitor cells and murine striatal models of HD [209].

##### Parkinson’s Disease (PD)

An important histopathological finding in PD is the earlier and more severe degeneration of noradrenergic neurons in the locus coeruleus than in dopaminergic neurons of the substantia nigra, which are more prone to oxidative damage. It was recently observed that noradrenergic and dopaminergic cell lines, as well as primary neuronal cultures from the rat locus coeruleus and ventral mesencephalon, responded differently to hydrogen peroxide oxidative stress with increased protein levels of phosphorylated p53 and γH2AX in the noradrenergic cells, indicating that these neurons are more vulnerable to DNA damage [210]. The activation of the DDR in vitro and in vivo was also validated in two synucleinopathy models of PD disease, leading to the demonstration that DNA damage was accompanied by oxidative stress and mitochondrial dysfunction [211].

##### Fragile X Syndrome

Fragile X syndrome is a common form of inherited intellectual disability caused by loss of the fragile X mental retardation protein (FMRP). FMRP is present predominantly in the cytoplasm, where it regulates the translation of proteins that are important for synaptic function. In addition, FMRP is a chromatin-binding protein that operates in the DDR. Interestingly, the loss of FMRP compromises the phosphorylation of H2AX in response to replication stress. Remarkably, hippocampal primary neurons from *Frmr1* knockout mice displayed an excessive internalization of AMPA receptors at their dendrites with a functional deficit in glutamatergic signaling [212].

##### Amyotrophic Lateral Sclerosis (ALS) and Frontotemporal Dementia (FTD)

ALS is a rapidly progressing neurodegenerative disease affecting motor neurons. Hexanucleotide (GGGGCC) repeat expansions in a non-coding region of C9orf72 are the major cause of familial ALS and FTD worldwide [213]. In clinical subtypes of both conditions, there is an abnormal cytoplasmic accumulation of the fused in sarcoma (FUS) protein in neurons [214]. FUS is a member of the FET family of DNA- and RNA-binding proteins that are involved in ALS, FTD, and other neurodegenerative diseases [215]. In the context of this contribution, it is of interest to recall that induction of DNA damage with the antibiotic calicheamicin γ1, which causes DNA DSBs, replicates several pathologic hallmarks of FTD-FUS in immortalized human cells, primary neurons, and astrocytes with the appearance of γH2AX nuclear foci [216]. Remarkably, elevated levels of γH2AX were detected in FTD-FUS brains, indicating that DNA damage also occurred in patients [216].

##### Others

Very rare polymorphisms in the human vaccinia-related kinase 1 (VRK1) gene have been identified in complex neuromotor phenotypes associated with spinal muscular atrophy (SMA), pontocerebellar hypoplasia (PCH), microcephaly, ALS, and distal motor neuron dysfunctions. The mechanisms by which VRK1 variant proteins contribute to the pathogenesis of these neurological syndromes are unknown. Molecular modeling predicted that VRK1 variant proteins are either unstable or have altered kinase activity. Among the substrates of the latter are γH2AX, HH3, and 53BP1, and, specifically, the G135 variant caused a defective formation of 53BP1 foci in response to DNA damage, suggesting that alterations in the DDR may be important factors in the onset of these pathologies [217].

Senataxin (SETX) is a DNA/RNA helicase critical for neuron survival [218]. SETX mutations cause two inherited neurodegenerative diseases: ataxia with oculomotor apraxia type 2 (AOA2) and ALS type 4 (ALS4). SETX is localized to distinct foci during the S-phase of the cell cycle, and these foci represent sites of DNA polymerase/RNA polymerase II collision, as they co-localize with the DNA damage markers 53BP1 and γH2AX, indicating an intervention of the DDR machinery in the two neurological conditions [219].

## 4. Conclusions

The phosphorylated form of H2AX has been initially related to the DDR in cells subjected to some sort of DNA damage. Then, many papers highlighted that γH2AX could not be merely considered a specific DNA DSB marker with a role limited to the DDR, but rather a molecule with pleiotropic functions. In this review, we have described these functions with particular attention to the role of γH2AX in the undamaged brain. The intervention of H2AX in processes, such as cell division and apoptosis, was also discussed to clarify some of the mechanisms by which the histone plays a remarkable function in certain very important neuropathological conditions.

## Figures and Tables

**Figure 1 molecules-26-07198-f001:**
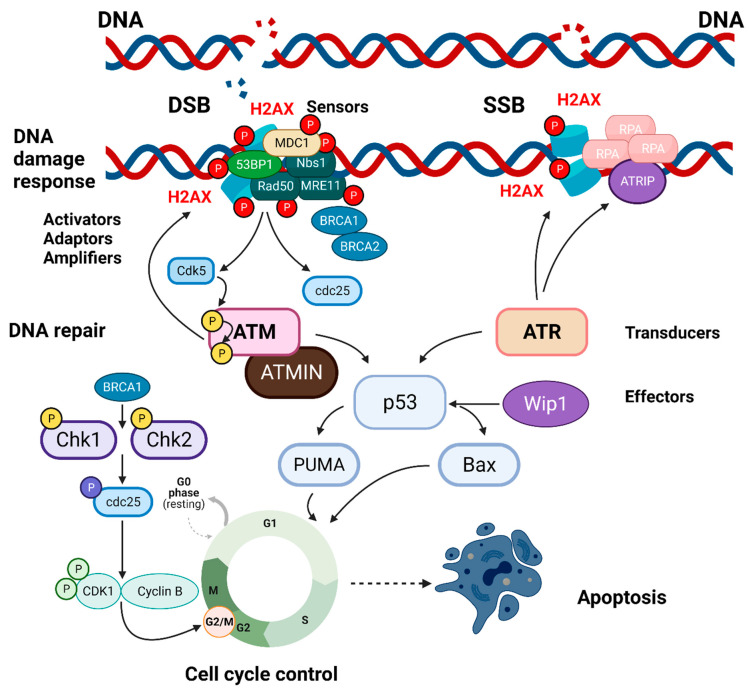
Simplified pathways of the DDR and cellular responses in which there is the participation of γH2AX. H2AX has a primary role in the repair of DNA DSBs (left), but it also intervenes in the mending of SSBs (right). This scheme also takes into consideration the intervention of γH2AX in the G2/M checkpoint. The response to DSBs starts with the focal accumulation of a series of sensor proteins, including MDC1, 53BP1, and the MRN complex (MRE11, Rad50, Nbs1). The phosphorylation of H2AX activates some transducer proteins, ATM being the most important. At the site of DNA damage, there is also an accumulation of BRCA1 and 2. The former activates the G2/M checkpoint that regulates the progression of the cell cycle from the G2 to the M phase, eventually leading to apoptosis when damage cannot be repaired. Repair of SSBs requires the intervention of a different set of proteins, including RPA, a highly conserved eukaryotic ssDNA-binding protein that is essential for genome stability. RPA interacts with ssDNA and with protein partners to coordinate DNA replication, repair, and recombination. Another important protein for SSB repair is ATRIP. ATRIP binds to ATR and to RPA-single-stranded DNA to drive ATR activation and thus facilitate recovery from replication stress. Created with BioRender.com.

**Figure 2 molecules-26-07198-f002:**
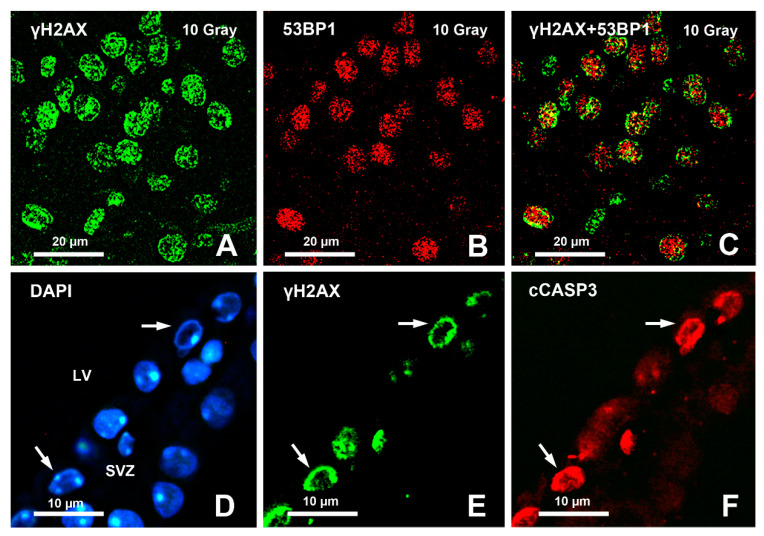
The pattern of nuclear γH2AX immunostaining after irradiation with X-rays of the old (24 months) mouse cerebral cortex (**A**–**C**) and in the SVZ of normal untreated old mice (**D**–**F**). Note that both γH2AX (**A**) and 53BP1 (**B**) display a focal pattern of staining. Panel (**C**) shows the merge of the two previous figures. Note that there are many double-immunostained foci (yellow) but also individual foci of γH2AX (green) or 53BP1 (red) immunoreactivity. In untreated mice, some SVZ cells (arrows) display a typical apoptotic ring of immunoreactivity at the periphery of the nucleus (**E**). Note that these cells are also immunopositive for cCASP3.

**Figure 3 molecules-26-07198-f003:**
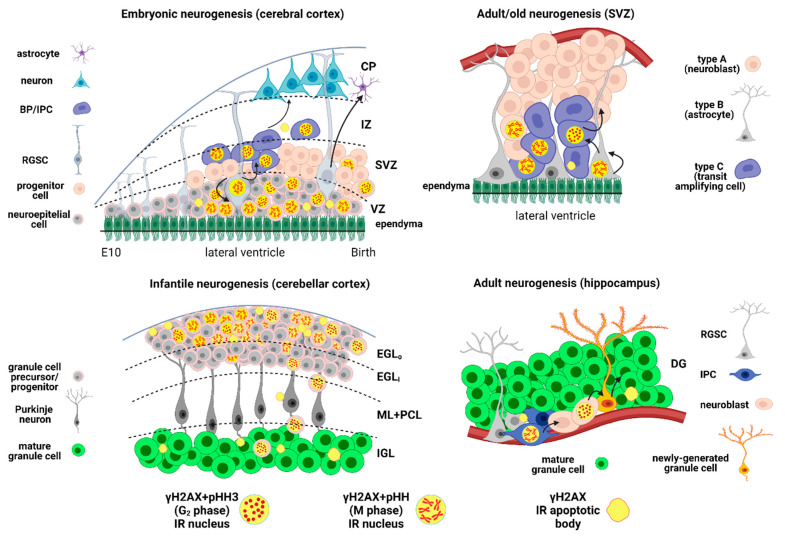
Simplified graphical summary of the immunocytochemical localization of γH2AX in different phases of neurogenesis in the mouse. Phosphorylation of H2AX mainly occurs in precursor/progenitor cells and RGSCs during the pre-mitotic G_2_ phase and mitotic M phase. Note that γH2AX is also detected in nuclei with an apoptotic morphology and in apoptotic bodies. In the cerebral cortex, the differentiation into RGSCs is followed by an asymmetrical division that gives rise to another RGSC and a more differentiated daughter cell referred to as abventricular or basal progenitor (BP) or intermediate progenitor cell (IPC). BPs migrate away from the apical progenitor domain and initiate neuronal differentiation. See text for further explanations. Abbreviations: BP = basal progenitor; CP = cortical plate; DG = dentate gyrus; EGL_i_ = inner (premigratory) layer of the external granular layer; EGL_o_ = outer (proliferative) layer of the external granular layer; IGL = inner granular layer; IPC = intermediate progenitor cell; IR = immunoreactive; IZ = intermediate zone; ML = molecular layer; PCL = Purkinje cell layer; RGSC = radial glia stem cell; SVZ = subventricular zone; VZ = ventricular zone.

**Figure 4 molecules-26-07198-f004:**
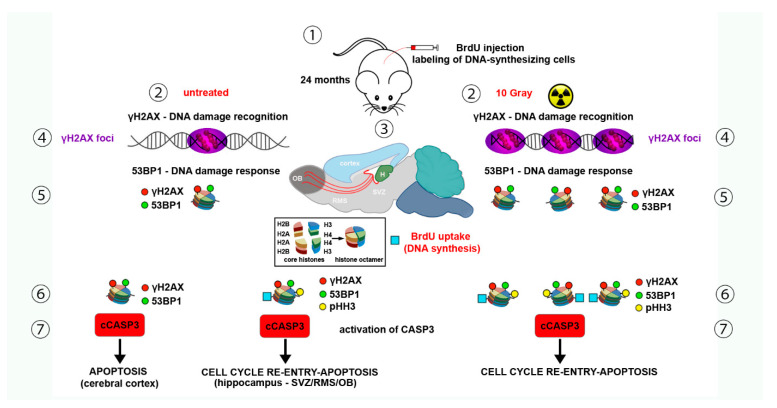
Occurrence of γH2AX in the forebrain of untreated and X-ray irradiated mice. ① To study the relationship between histone H2AX phosphorylation and DNA synthesis, mice were injected with BrdU and left to survive for two hours. ② Animals were untreated or irradiated with a 10 Gray dose of X-rays. ③ Brains were sectioned and processed for the immunocytochemical detection of γH2AX, 53BP1, pHH3, BrdU, and cCASP3 in the cerebral cortex, hippocampus, and SVZ/RMS/OB. ④ γH2AX immunoreactive foci and ⑤ 53BP1 immunoreactive foci are detected in both untreated and X-ray irradiated mice, albeit at a much higher extent after X-ray irradiation. ⑥ BrdU is incorporated in newly synthesized DNA except in post-mitotic cortical neurons of untreated mice. DNA synthesis is related to the initiation of an aberrant cell cycle, with phosphorylation of HH3. ⑦ Activation of caspase 3 and apoptosis occur directly without cell cycle re-entry in the cerebral cortex. For details about the experimental procedure, see Figure 1 in [117]. Abbreviations: BrdU = 5-Bromo-2′-deoxyuridine; cCASP3 = cleaved caspase 3; H = hippocampus; OB = olfactory bulb; SVZ = subventricular zone; RMS = rostral migratory stream.

**Table 1 molecules-26-07198-t001:** Neurotoxic substances that induce γ phosphorylation of H2AX in the nervous system. Abbreviations: Ara-C = Cytosine arabinoside; ATM = ataxia telangiectasia mutated; cCASP3 = cleaved caspase 3; CGRP = calcitonin gene-related peptide; CHK2 = Checkpoint kinase 2; DG = dentate gyrus; DSP4 = N-(2-chloroethyl)-N-ethyl-2-bromobenzylamine; FANCD2 = Fanconi anemia D2 protein; MDC1 = Mediator of DNA damage checkpoint protein 1; TCDD = 2, 3, 7, 8-tetrachlorodibenzo-P-dioxin.

Chemical/Drug	Experimental Target	Mechanism of Action	Other Effects besides Induction of γH2AX	References
Actinomycin D	Rat sensory ganglion neurons	Inhibition RNA synthesis	Heterochromatin silencing	[184]
Ethanol	Mouse brain and human neuronal cells	Induction of apoptosis	Induction of FANCD2	[185]
Camptothecin	Rat cortical neurons	Inhibition of topoisomerase I with apoptosis	Activation of ATM, CHK2, MDC1, and 53BP1	[186]
Temozolomide	Human glioblastoma cell lines	Methylation of DNA guanine bases with apoptosis	N/A	[187]
Mifepristone	Mouse photoreceptors	Glucocorticoid receptors antagonism	Induction of pro-apoptotic factors	[188]
DSP4	SHSY5Y cells	Block of noradrenaline uptake	Degeneration of noradrenergic terminals	[189]
TCDD	SHSY5Y and PC12 cells	Activation of the aryl hydrocarbon receptor	Premature senescence	[190]
Cisplatin, oxaliplatin, carboplatin	Cultured rat sensory neurons	Crosslink with the DNA urine bases with apoptosis	Reduction of the capsaicin-evoked release of CGRP	[191]
Ara-C	Cultured mouse hippocampal neurons	Inhibition of DNA polymerases, block of cell mitosis	N/A	[192]
Cypermethrin	Adult zebrafish retinal cells	Disruption of voltage-gated Na^+^ channel function	Increase of cCASP3	[193]
Zinc oxide nanoparticles	SHSY5Y cells	Viability decrease, apoptosis, cell cycle alterations DNA damage	Production of micronuclei	[194]

**Table 2 molecules-26-07198-t002:** Oxidative stress induction of γ phosphorylation of H2AX in the nervous system. Abbreviations: 8-OHdG = 8-hydroxy-2′ –deoxyguanosine; ERCC1 = ERCC excision repair 1, endonuclease non-catalytic subunit; KA = kainic acid; Mre11 = MRE11 homolog, double strand break repair nuclease; NADPH = reduced nicotinamide adenine dinucleotide phosphate; NMDA = N-methyl-D-aspartate; RAD50 = RAD50 double strand break repair protein; ROS = reactive oxygen species; TRESK = TWIK-related spinal cord K^+^ channel.

Oxidative Stress Inducer	Experimental Target	Mechanism of Action	Other Effects besides Induction of γH2AX	References
Fluorescent immunohistochemical techniques	Cultured rat cortical neurons	Supra threshold activation of ionotropic glutamate receptors	Induction of MRE11	[124]
KA	Rat hippocampus and entorhinal cortex in vivo	Activation of KA receptors and induction of seizures	Induction of MRE11	[195]
NMDA	Mouse retinal ganglion cells and inner nuclear layer cells	Activation of NMDA receptors	Increase in 8-OHdG and TUNEL positive cells	[196]
Hydrogen peroxide	BE(2)C neuroblastoma cells	Induction of oxidative stress	Change in cellular levels of MRE11, RAD50, nibrin, and ERCC1	[197]
Genetic mutation	Glucose-6-phosphate dehydrogenase-deficient mice	Reduction in NADPH levels	Synaptic and behavioral disorders	[198]
Sevoflurane	In vitro and in vivo rat neurons	Decrease of gap junction mediated cell-cell coupling and alteration of the action potential	Increase of intracellular ROSNeuronal cell parthanatos	[199]
TRESK silencing	Cultured mouse spinal cord dorsal horn neurons	Regulation of primary sensory neurons excitability	Induction of apoptosis	[200]
Sterigmatocystin	Rat hippocampal DG	Induction of oxidative stress, mitochondrial dysfunction, apoptosis, cell cycle arrest	Disruption of postnatal neurogenesis and adult-stage suppression of synaptic plasticity	[201]

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
