# Peer review of "The Phosphorylated Form of the Histone H2AX (γH2AX) in the Brain from Embryonic Life to Old Age"

_molecules, 2021, doi:10.3390/molecules26237198_

Round 1

Reviewer 1 Report

  1. The review is very long. I suggest the authors should trim it to be more attractive.
  2. Line: 276-287: If H2AX is related to mitochonria, you only focus on brain. In fact, mitochondria is distributed at many areas, such as brain, heart, muscle......... Is any relation between the H2Ax and heart disease? muscle disease?
  3. Page 18. I suggest you to add a list including different part of CNS, like brain, cerebraum, hippocampus.....at different stage (perinatal, postnatal, infant, and adult) to help your readers understanding the role of H2Ax in the neurogenesis.

Author Response

Thank you very much for the appreciation of our work. Please find below the responses to your observations/requests.

  1. We have shortened the paper as much as possible.
  2. Thank you for this observation. We have been unable to find any direct correlation between γH2AX and mitochondrial homeostasis in current literature except for ref 67. As mentioned in the original MS the effect of a lack of H2AX was related to a loss of PGC-1α with an impairment of the respiratory chain in the brain and cultured fibroblasts. Such an observation is not described in other tissues to the best of our knowledge. Yet it seems reasonable that, if investigated, a role of H2AX in mitochondrial homeostasis will be found also in other tissues. We have mentioned this in the revised MS (lines 293-294).

  3. Thank you very much for this additional very useful comment. In response, we added a new figure (Figure 3 in the revised MS) that resumes graphically the data on the role of ΥH2AX in neurogenesis.

Reviewer 2 Report

The review entitled “ϒ-H2AX in the brain from embryonic life to old age” by Merighi and collaborators, provides an overview on the functions of the phosphorylated form of histone H2AX in the brain, covering aspects of the role of this histone variant in various cellular processes. After that, they have reviewed the expression and the possible functions of this histone in different brain areas, under normal and pathological conditions.

The review is well structured and appears to contain several citations necessary to cover all the above aspects. Therefore, it may provide a comprehensive story, useful for scientist working on ϒ-H2AX with an interest focused on the brain.

However, in general, the review is too long, and in certain parts the details are too many, resembling the description of a research article, rather than providing a summary useful to understand, and compare the relevance of the described findings with other tissues/organs/conditions.

For instance, the section 2.3.1 and parts therein, reports several details of the experimental results (see lines 711-728, p15-16; lines 733-754, p. 16), making it heavy the reading of the review, which should present more concisely the matter to obtain the relevant information.  

Similarly, Figure 3 anticipates the discussion of H2AX  should be ϒ-H2AX) and its relationship with apoptosis in the old mouse brain, but a previous explanation would be useful, even for understanding the figure itself (whose legend is also not very clear).

Other points:

Title: it refers to ϒ-H2AX (please add the word “histone”), while several headings and sub-headings report only H2AX.  Specify which form is referred to in each section.

Figure 2: Pictures showing untreated controls are missing. Please, provides relevant images.

Section 1.10 H2AX and tumors: here, it should be described the occurrence (if any) of ϒ-H2AX in brain tumors.

Author Response

Thank you very much for the appreciation of our work. Please find below the responses to your observations/requests.

In response to the general comment about the length of the paper and specific observations regarding section 2.3.1 we have shortened the paper as much as possible. In particular we have cut the parts in which our previous reserach was describedm, referring to published data.

In response to comment about Figure 3 in the original MS (now Figure 4) we would like to first thank the reviewer for the very useful observations. Accordingly, we have modified the figure to make it more readable and corrected the legend with a point-to-point description of the different steps described in the figure workflow.

We have changed the title and checked headings and subheadings for correctness in indicating H2AX or γH2AX.

As regarding the comment on Figure 2, we respectfully disagree with as panels D-F indeed referred to the SVZ of control animals. Indeed we have not shown controls for the cerebral cortex, but this figure has only an exemplification meaning and it is not intended to show original results that have been already published in ref 117.

About the occurrence of γH2AX in brain tumors, it must be said that it has been described mainly about the response of chemotherapy treatments and not as a primary intervention of the histone in tumorigenesis. Therefore we have not considered it thoroughly in this section. Yet we have mentioned the brain tumors in which H2AX phosphorylation was also reported concerning the onset of a dysregulation of the DDR machinery.